# miR-430 microRNA Family in Fishes: Molecular Characterization and Evolution

**DOI:** 10.3390/ani13152399

**Published:** 2023-07-25

**Authors:** Claudio A. Jiménez-Ruiz, Roberto de la Herrán, Francisca Robles, Rafael Navajas-Pérez, Ismael Cross, Laureana Rebordinos, Carmelo Ruiz-Rejón

**Affiliations:** 1Departamento de Genética, Facultad de Ciencias, Universidad de Granada, Avda. Fuentenueva s/n, 18071 Granada, Spain; 2Área de Genética, Facultad de Ciencias del Mar y Ambientales, Instituto Universitario de Investigación Marina (INMAR), Universidad de Cádiz, 11510 Cádiz, Spain

**Keywords:** noncoding RNA, tandem repeats, genomic organization, phylogenetic and evolutionary analysis

## Abstract

**Simple Summary:**

MicroRNAs regulate gene expression and, in particular, the miR-430 family plays an important role in the early development of fishes. In this study we have observed that this family appeared early in the evolution of fishes and all the studied species have multiple copies of miR-430. In some fishes that diverged early on, we have found three different versions of miR-430, yet as fishes evolved over time, some groups seem to have lost some of these versions. These findings could lead to more studies on the different functions of these versions, which could be preserved in farmed species. This knowledge could help improve aquaculture methods.

**Abstract:**

The miR-430 microRNA family has been described in multiple fish species as one of the first microRNAs expressed by the zygote. It has been suggested that this family is implicated in maternal mRNA elimination, but may also play a role in steroidogenesis, sexual differentiation, and flatfish metamorphosis. The miR-430 sequences have been found in multiple-copy tandem clusters but evidence of their conservation outside of teleost fishes is scarce. In the present study, we have characterized the tandem repeats organization of these microRNAs in different fish species, both model and of interest in aquaculture. A phylogenetic analysis of this family has allowed us to identify that the miR-430 duplication, which took place before the Chondrostei and Neopterygii groups’ divergence, has resulted in three variants (“a”, “b”, and “c”). According to our data, variant “b” is the most closely related to the ancestral sequence. Furthermore, we have detected isolated instances of the miR-430 repeat subunit in some species, which suggests that this microRNA family may be affected by DNA rearrangements. This study provides new data about the abundance, variability, and organization of the miR-430 family in fishes.

## 1. Introduction

MicroRNAs (miRNA) are small noncoding RNAs (~22 nt) that play a crucial role in post-transcriptional gene regulation. They generally achieve this by binding of the mature miRNAs to the 3′ untranslated region of target messenger RNAs (mRNA) and promoting their degradation. Animal miRNAs primarily recognize their targets through a seed sequence located between the second and eighth position of the miRNA’s 5′ end [1]. Due to their functional importance, miRNAs are under selective pressure and are conserved even amongst distantly related species, making them important when investigating the evolution of species. Moreover, miRNAs exhibit dynamic evolutionary changes, including gene duplication, gain or loss of target sites, and acquisition of new functions. In this sense, miRNAs are often duplicated, forming clusters that can be transcribed into polycistronic transcripts and maintain physical linkages even after long evolutionary periods [2,3].

The miR-430 family has been described in fishes and is orthologous with identical seed sequences in amphibians and mammals [4]. This family is involved in various biological processes of interest in the field of aquaculture, including early development, left-right asymmetry, flatfish metamorphosis [5,6], and regulation of steroid production [7]. However, it has not been widely studied in fish species farmed in aquaculture or with interest in fisheries. The miR-430 family was first described in zebrafish (*Danio rerio*) [8], and stands out for being among the first and most abundantly expressed genes during the zygotic genome activation [9,10] when it promotes the degradation of the maternal mRNAs [11,12]. This clearance of maternal mRNAs is required for maternal–zygotic transition [12] and promotes the establishment of global heterochromatin [13]. The miR-430 family is also required for the establishment of the embryonic body plan [12]. The studies involving zebrafish with disrupted production of miR-430 have found a mismigration of primordial germ cells [14], defects in multiple organs [8,15], and a developmental delay and lethal phenotypes [12]. MiR-430 is also needed to suppress the expression of primordial specific genes in somatic cells, a function that might be conserved in *Oryzias latipes* [16]. In zebrafish, this family includes several variants named from “a” to “j” [8,10]. Variants “a”, “b” and “c” are the most highly expressed [8], and have homologous miRNAs in other species [5,16,17]. For instance, in *Monopterus albus*, these variants show tissue and sex-specific expression and may be involved in sex reversal [7,17], while in the *Paralichthys olivaceus* variant, “a” plays a role in metamorphosis and regulates the left-right asymmetry [5,6].

Regarding the organization of the miR-430 family, in zebrafish it was first described that there were two clusters in chromosome four formed by multiple copies of the different variants [18]. More recently, it has been reported that in zebrafish there is a cluster in chromosome four formed by tandem repeats of triplets of the variants “a”, “b” and “c” with a promoter sequence every two or three triplets. Thus, this locus has the greatest promoter density in the entire genome of the zebrafish, which produces an active transcription body that could mediate the transcription of other genes [19]. Multiple copies of miR-430 have been identified in other fish species, such as *Takifugu rubripes*, *Tetraodon nigroviridis*, *O. latipes*, and *Carassius auratus* [8,10,16,19]. These findings have led to the hypothesis that the teleost ancestor likely presented tandem repeats containing multiple copies of miR-430 [16]. The organization in triplets, however, seems to be absent in other species [10,19]. Moreover, no transcript homologous to miR-430b has been found in *T. rubripes* or *O. latipes*, so it has been hypothesized that this variant has been acquired independently during the teleost evolution [16].

This study aims to: (1) test the mode of evolution of the miR-430 family and its variants in jawless, cartilaginous and bony fishes, including economically relevant fish species; (2) investigate the interrelation of these variants, shedding light on their evolutionary history; and (3) examine the conservation of the tandem organization and the high promoter density of the miR-430 loci throughout fish evolution.

## 2. Materials and Methods

### 2.1. Species Selection

For this study, we selected fish species based on the availability of randomly selected genomic paired-end Illumina reads that could be used in the RepeatExplorer2 pipeline, as described in the next section, and chromosome-level genome assemblies. Although the *Callorhinchus milii* genome was only at the scaffold level, we used it as a representative of the Chondrichthyes class, in which no other species met the previous criteria. The three resulting groups were:(A)Well-studied/model organisms: *D. rerio*, *Gasterosteus aculeatus*, *O. latipes*, *Petromyzon marinus*, *T. rubripes*, and *T. nigroviridis*, plus some species of the genus *Danio* as an example of closely related species: *Danio aesculapii*, *Danio albolineatus*, and *Danio nigrofasciatus*.(B)Species farmed in aquaculture and/or with interest to fisheries: *Acipenser ruthenus*, *Amia calva*, *Atractosteus spatula*, *C. milii*, *Gadus morhua*, *Micropterus salmoides*, *Salmo salar*, *Salmo trutta*, and *Solea senegalensis*.(C)Cichlids species, with more variability in the number of clusters and the seed sequence: *Amphilophus citrinellus*, *Astatotilapia stappersii*, *Ctenochromis horei*, *Cyprichromis leptosome*, *Haplochromis burtoni*, *Labeotropheus trewavasae*, *Melanochromis auratus*, *Oreochromis korogwe*, *Oreochromis niloticus*, *Perissodus microlepis*, *Pseudocrenilabrus philander*, and *Pundamilia nyererei*.

### 2.2. Characterization of the miR-430 Locus

#### 2.2.1. RepeatExplorer2 Analysis

For each species, genomic paired-end Illumina reads were downloaded from the Sequence Read Archive (SRA) (see Appendix A); only sequencings with a low proportion of overlapping read pairs were used. The quality of the raw forward and reverse reads was checked with FastQC v0.11.2 (https://github.com/s-andrews/FastQC, accessed on 1 February 2023). The reads were quality filtered, cleaned from adapters, and interlaced using the “Pre-processing of FASTQ paired-end reads” tool on the RepeatExplorer Galaxy portal. All the sequencings presented uniform base contents after filtering. The filtered reads were used to identify clusters of repetitive DNA using the RepeatExplorer2 pipeline [20] with default parameters but using the Metazoan version 3.0 REXdb. This pipeline identifies tandem repeats without the need of a reference genome thanks to the graph-based clustering of reads and then classifies these clusters in different types of repetitive elements. RepeatExplorer2 includes the output from the TAREAN pipeline [21], which identifies satellite repeats based on the graph structure of the cluster. For each species, the contigs of the detected clusters were used for a BLASTn search [22] using the stem-loop sequences corresponding to miR-430 from miRBase [23] to identify the miR-430 clusters. Hereafter, when referring to the entire hairpin that contains the miR-430, we will refer to it as the miR-430 sequence; when referring to the tandem repeat loci where the miR-430 sequences are located, we will use the term miR-430 cluster; and consensus repeat subunit will be used when referring to the consensus of the tandem repeats. The consensus repeat subunit was obtained from the TAREAN output, but in the cases of smaller clusters, that were not classified as satellite clusters, the biggest contig was used as the repeat subunit. The percentage of the genome occupied by miR-430 clusters was calculated as the proportion of the reads that formed the cluster against the nuclear analysed reads. For *Danio* ssp. the promoter sequences were detected in the different consensus repeat subunits by BLASTn search against the previously described promoter of *D. rerio* [24]. In the rest of the species, this analysis was performed using the Neural Network Promoter Prediction tool (https://fruitfly.org/seq_tools/promoter.html, accessed on accessed on 15 February 2023).

#### 2.2.2. Genomic Location

Assembled genomes at the chromosome level were downloaded from the NCBI database. MiR-430 clusters were identified by BLASTn using the clusters detected with RepeatExplorer2. The identified regions were inspected using Geneious v2022.0.2, which was also used to construct dot plots to spot repetitions. Tandem Repeats Finder [25] was used to detect the consensus repeat subunit of *C. milii* in the scaffolds with homology to miR-430. When isolated repeat subunits were detected, the presence of upstream and downstream mobile or repetitive elements was investigated. For *D. rerio* and *D. aesculapii* previously described transposable elements were detected using the Dfam web tools [26] with the *D. rerio* data, while for *S. salar* the contigs obtained with RepeatExplorer2 were used to detect undescribed mobile elements with BLAST.

### 2.3. Identification and Analysis of miR-430 Sequences

Putative miR-430 sequences were first identified in the consensus repeat subunit by similarity to previously described miR-430 sequences and the presence of the seed sequence. Then, the secondary structure of the region that included the seed sequence and 100 nucleotides upstream and downstream was analysed using the RNAfold web server (http://rna.tbi.univie.ac.at/cgi-bin/RNAWebSuite/RNAfold.cgi, accessed on 2 March 2023). The largest stable hairpin structures that did not include other hairpins were selected as miR-430 sequences. Two phylogenetic trees of miR-430 sequences were constructed, one with model species and another one with species of interest to fisheries and aquaculture. These phylogenetic trees were produced using the maximum-likelihood method with the Tamura’s 3-parameter model with gamma distribution and invariable sites as the best evolutionary model proposed by MEGA11 [27] over a Clustal Omega [28] alignment. The robustness of the tree was verified by 1000 bootstraps pseudo-replicates. Once the conservation of the different variants was established, miR-430 sequences were classified considering their similarity to the mature miR-430 variants of *D. rerio*.

## 3. Results

### 3.1. Organization

From the repetitive DNA clusters of *S. senegalensis* described in a previous article [29] we detected that the consensus repeat subunit of cluster 31 had an 86.96% percentage identity in relation to *O. latipes* miR-430d. Figure 1 exemplifies the miR-430 organization found in *S. senegalensis*. In the case of *S. senegalensis* the miR-430 cluster covers 0.048% of the genome and forms a single locus in chromosome five, and the consensus repeat subunit includes two miR-430 sequences and a putative promoter sequence. In the rest of the species, excluding the cichlid species, the miR-430 clusters comprise between 0.0119% and 0.085% of their respective genomes, with consensus repeat subunits ranging from 251 to 1443 base pairs (bp) (Table 1). The analysis of the consensus repeat subunit of each species resulted in the identification of the number of miR-430 sequences per repeat subunit, which ranged from one to four. The resulting miR-430 sequences grouped with the miR-430 “a”, “b” and “c” variants described in *D. rerio*, as shown in the alignment in Appendix A, and were therefore classified based on their similarity to these variants. The presence of a promoter sequence was also examined in each of the consensus repeat subunits. The miR-430 promoter previously described in *D. rerio* was not detected in species outside of the genus *Danio*. Nevertheless, other putative promoters were detected in the rest of the species except for *P. marinus*, *M. salmoides*, and *G. morhua*.

### 3.2. Genomic Location

The genomic location and the number of loci of the miR-430 clusters are included in Appendix A, all the clusters spanned more than 6 kilobases (kb). Most of the studied species presented the miR-430 cluster in a single locus, while *S. salar* and *S. trutta* presented tandemly repeated clusters in two chromosomes, both detected by RepeatExplorer2 as a single cluster. Only the genomes of *O. niloticus* and *O. latipes* had two clusters in the same chromosome separated by more than 10 kb. In the case of the genome assemblies of *D. aesculapii*, *D. rerio*, and *S. salar*, apart from chromosomes that presented miR-430 clusters (Appendix A), we found isolated instances of the repeat subunit in other chromosomes (*D. rerio* and *D. aesculapii* chr10 and *S. salar* chr19 and chr26). In the case of *D. rerio*, there is a nonautonomous DNA transposable element (DF0002749.1 [MuDR-N1_DR]) inserted in the region that would correspond to the miR-430b gene. This isolated repeat is flanked by a satellite element (DF0003427.1 [SAT-12_DR]) and another nonautonomous DNA transposable element (DF0002813.1 [hAT-N53B_DR]), which are 112 bp and 6 bp away from the repeat subunit, respectively (Appendix A). In *D. aesculapii*, the chromosome 10 miR-430b sequence is also truncated and flanked by a nonautonomous DNA transposable element repeat (DF0002824.1 [DNA-1-9_DR]) 516 bp away and by a LINE (DF0002485.1 [REX1-5_DR]) 105 bp away. In chromosome 19 of *S. salar*, the isolated repeat is linked to a transposable element Tcb1 and 274 bp away from another Tcb1 element. While in chromosome 26, there is only one unclassified repetitive element 483 bp away from the isolated repeat. In the three species, the pairwise distance was greater when comparing the isolated miR-430 sequence of each species than when the isolated sequences were compared to the clustered miR-430 sequences of the same species.

### 3.3. Evolution

The comparison of the consensus repeat subunits of four *Danio* species resulted in a mean interspecific identity of 76.8%, while the miR-430 sequences had a pairwise identity of 89.7% and the promoter sequences 79.5%. The mean interspecific identity of the repeat subunit between *S. salar* and *S. trutta* was 95%, while the miR-430 sequences had a pairwise identity of 98.9% and the promoter sequences 98.7%. However, when we tried to compare consensus repeat subunits of species from different genera, such as *T. rubripes* and *T. nigroviridis*, these were highly divergent due to the different number of miR-430 sequences, the variability in length and sequence of the spacers between them, and the lack of a specific order of the variants.

#### Phylogeny

In the phylogenetic analysis, we detected that variant “b” is more closely related to the miR-430 sequences found in *P. marinus* (jawless fish), used as an outgroup for being the most basal species. Variants “a” and “c” are more proximate between each other and more distant to variant “b”, in both cases whether we analyse model species (Figure 2a) or species of interest to fisheries (Figure 2b). In the last-mentioned analysis, the miR-430 sequence of *C. milii*, a cartilaginous fish as *P. marinus* groups closely with the one found in this last species. The variants “a”, “b” and “c” are nevertheless found in the Chondrostei species *A. ruthenus* and in Neopterygii species, all of them in the bony fish group. In several of these species; however, we could not detect all three variants in the consensus repeat subunit.

### 3.4. miR-430 Organization and Seed Sequence Variability in Cichlids

In all the 12 studied species of the Cichlidae family, we could not find miR-430 sequences with homology to variant “b”; however, we found miR-430 sequences with a mutation in the seed sequences, changing from AAGUGCU to AAGUGCA. We also detected stable hairpins with the canonical seed sequence in the 5′ arm of the hairpin in two species (*C. horei* and *H. burtoni*). Moreover, the species of this family show a wider variety of organizations, having between one and four different miR-430 clusters, each comprising less than 0.02% of the genome. Nonetheless, the number of clusters was the same within genera (Table 2). Only *O. niloticus* and *A. citrinellus* had both a genome assembly at the chromosome level and short reads available, and in both species, the number of clusters found was the same regardless of the analysed data. In the *O. niloticus* genome assembly the two clusters were separated by ~2–3 kb but in the same chromosome. In *A. citrinellus* and *A. stappersii,* only one miR-430 cluster has been detected and the miR-430 sequences identified in these clusters have the mutated seed sequence AAGUGCA. Thus, these two species are the only instances in which we have not found at least one miR-430 sequence that has the AAGUGCU seed sequence.

## 4. Discussion

In this study, we have analysed the conservation of the miR-430 family, finding that these miRNAs are maintained in all the studied fish species. This family had already been characterized in 10 fish species: *Astatotilapia burtoni*, *Cyprinus carpio*, *D. rerio*, *G. morhua*, *Hippoglossus hippoglossus*, *O. latipes*, *O. niloticus*, *P. marinus*, *P. nyererei* and *S. salar* (miRBase, accessed on 30 January 2023). A previous study suggests that this family might have been acquired during the craniate evolution, as it is absent in tunicates and cephalochordates [28]. This hypothesis correlates with the absence of this miRNA in species of other superphyla, such as insects and molluscs.

Although there are many miR-430 variants described in *D. rerio* and other species [8,10,16,30,31], according to our results, the sequences from most species (except *P. marinus*, *C. milii* and the mutated sequences found in cichlids) can be clustered into three groups that correspond with variants “a”, “b” and “c” of *D. rerio*. Therefore, in this study we have used these variants to name the sequences and opt for a simplified nomenclature as individual differences can be due to variability between repeats. These three variants are common to species in the Chondrostei and Neopterygii lineages, while the miR-430 sequences found in *P. marinus* and *C. milii* do not group with the aforementioned variants. Thus, the ancestral duplication that led to these three variants may have occurred during the evolution of bony fishes or be specific to Actinopterygii, which could be due to the greater need for regulation of the bony fishes. Variant “b” would be the original, being the most closely related to the ancestral variant (Figure 2). However, a greater availability of genomic data will be needed to confirm this.

The tandem organization of the miR-430 sequences is also conserved from cartilaginous to bony fishes. All the miR-430 clusters found, excluding the ones in cichlid species, occupied a minimal percentage of the genome of ~0.01%, but this percentage was quite heterogeneous even inside the same genus (e.g., the genus *Danio*). We found no correlation between the size of the repeat subunit and the percentage of the genome occupied by the miR-430 cluster. The number of miR-430 sequences in the repeat subunit also does not correlate with the size of the repeat subunit or the size of the miR-430 cluster (see Table 1). Previously, this cluster organization has been associated in *D. rerio* with the formation of a transcription body organizer during the minor wave of zygotic genome activation [19]. Moreover, in the *Xenopus laevis* paralogous miR-427 it has already been hypothesized that the high copy number of these miRNAs is necessary for clearing the maternal mRNAs in the zygote [32]. In *Drosophila melanogaster*, nonetheless, miR-309 has a similar function clearing the maternal mRNAs in the zygote, but it does not have this tandem organization [33], so this repetitive organization may not be essential for its function. This repetitive organization may nevertheless influence the evolution of the miR-430 family in two aspects. First, it might allow variant loss, which seems to be a common phenomenon, with variant “b” being the most commonly lost in our data. Second, this organization may have resulted in the concerted evolution of the repeats, causing them to be more similar inside the same species than between different species. The results that we have found here for the consensus repeat subunits are similar to those of the mammalian family miR-290-295, which contains some miRNAs that share the same seed sequence as miR-430. The miR-290-295 family has been described as highly variable throughout its evolution; between species of different orders, only the miRNA hairpins and the putative minimal promoter are conserved, while the number of pre-miRNAs and their flanking regions are variable [34]. However, in the case of the miR-430 cluster, the transcription promoter sequences found in some species are not conserved even in closely related species. This can be caused by the high intergenomic divergence, making promoters degenerate and arise from the spacers between miRNAs. In some species, we were not able to detect a promoter sequence in the consensus subunit. In this species, the promoter could be located outside of the miR-430 cluster, but a lower promoter density may not be essential for the miRNA function.

In our study, we have found that there are tandem repeats usually in just one chromosome and one locus per chromosome. Nonetheless, there are some exceptions, such as both species of the *Salmo* genus that have miR-430 repetitive clusters in two chromosomes, which could be a result of the salmonid whole genome duplication. In addition, *D. rerio*, *D. aesculapii*, and *S. salar* present isolated repeats but in other chromosomes. In *D. rerio*, we have found an isolated repeat in chromosome 10, while previously a similar sequence had been described in chromosome 13 of *D. rerio* [10], probably due to the difference in assemblies. These isolated repeats could be a result of interchromosomal rearrangements following the teleost whole-genome duplication, as has been hypothesized for the two miR-430 clusters previously described in chromosome four of *D. rerio* [16,18]. However, we have found that the miR-430 sequences in these isolated repeats are more similar to the clustered miR-430 sequences of that species than to the isolated sequences of other species, which suggests that these repeats have isolated independently and more recently. Furthermore, we have found transposable elements near or inside the isolated repeats in these three species (Appendix A), which could have mediated the transposition of these repeats. Transposition and subsequently tandem clustering could have caused the loss of some variants in some species or clades via loss during transposition, as all the suspected cases of transposition that we have encountered are characterized by loss or truncation of at least one variant. This possibility agrees with a previous study that concluded that the miR-430 family appears to be mobile based on the lack of synteny between the miR-430 sequences [35]. Moreover, in humans, the miRNA cluster found on chromosome 19, which includes some miRNAs with the same seed sequence as miR-430, has been related to Alu elements, and these elements have mediated the expansion of this cluster [36].

In cichlids, apart from the miR-430 sequences with the canonical seed sequence, we found miR-430 sequences with a mutated seed sequence in the 8th position of the mature miRNA. The origin of this mutation seems to be previous to the divergence of the African and American populations as species found in both continents share the presence of this seed sequence. Cichlids are widely studied for their rapid divergence and speciation and they present lake-specific or even species-specific miRNAs, which could play an important role in this rapid evolution [37]. This could explain their greater heterogeneity in the number of miR-430 clusters and the mutation of a seed sequence that is conserved in every other studied fish species. It should be noted, though, that the selection against mutations in the seed sequence’s 8th position may be weaker than in the other positions. As, for example, according to the TargetScanFish server (https://targetscan.org/fish_62/, accessed on 5 March 2023) in *D. rerio* multiple targets of the miR-430 family form 8mers with the seed sequence but most form 2–7mers. In fact, referring to mammals, a similar mutation may have occurred in the aforementioned miR-290-295 family, in which some miRNAs have the seed sequence AAGUGCC. Subsequent studies would be needed to establish the changes in target recognition of these mutated miRNAs.

Moreover, in this study, we have used RepeatExplorer2 to identify the genomic repetitive elements and later locate the miR-430 cluster among them by BLASTn search. RepeatExplorer2 has the advantage of being able to find repetitive elements without the need for a genome assembly and allows the detection of elements that could be under-represented in the assembly. Nevertheless, it has limitations; for example, nonrandom library preparations could result in sequencing bias and therefore over-representation of certain repetitive elements [20]. To avoid this, we have used only randomly selected sequencings that did not show over-represented sequences or contaminants in the FastQC analysis. However, other limitations are more difficult to avoid, such as failure to detect similarities between the reads and reduced sensitivity to less frequent or highly variable elements [20]. This could cause greater variability in the number of miR-430 clusters that we observe in cichlids, as it is possible that some of them are not detected because they occur less frequently than the miR-430 clusters of the other species. Other analyses and a greater availability of genome assemblies would be needed to address these issues.

## 5. Conclusions

In this study, we have characterized the miR-430 family in fishes, which is involved in the early development by clearing the maternal mRNAs. The sequences of this family are organized in cluster repeats and are usually located in a single chromosome. The comparative analysis of these clusters shows that this family evolves through concerted evolution. The repetitive subunit includes multiple miR-430 sequences that in most cases can be grouped into three variants (“a”, “b” and “c”). The duplication that led to these variants occurred during the evolution of bony fishes or Actinopterygii, although some species have lost at least one of these variants. In cichlids there are miR-430 clusters containing a mutated seed sequence and further studies should check the biological function of these miRNAs. This study illustrates the evolution of a miRNA family of interest to aquaculture due to its fundamental role in the early development of fishes.

## Figures and Tables

**Figure 1 animals-13-02399-f001:**
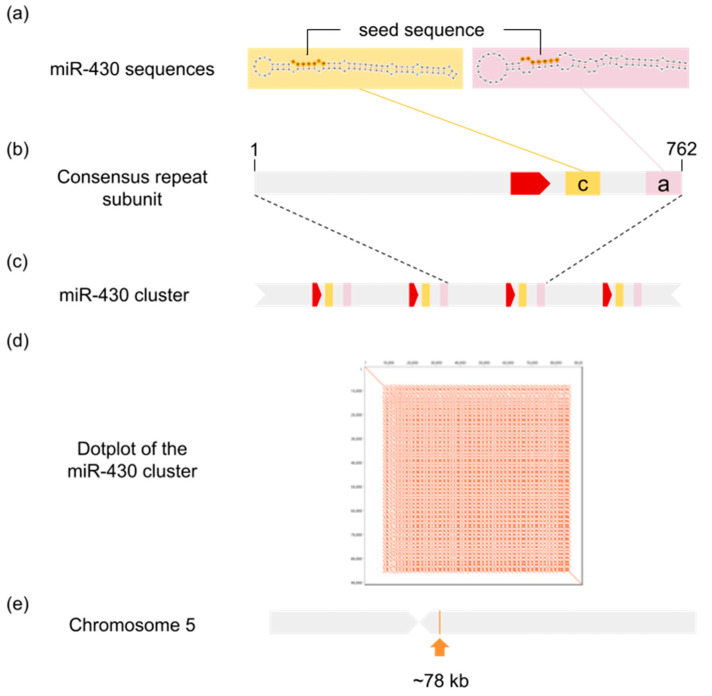
Organization of miR-430 sequences in *Solea senegalensis*. (**a**) Secondary structure of the miR-430 sequences, variant “c” coloured in yellow and variant “a” in pink, the seed sequence has been highlighted. (**b**) Representation of the consensus repeat subunit in which the miR-430 sequences and the promoter sequence (red) have been identified. (**c**) Example of the tandem organization of the miR-430 cluster. (**d**) Dot plot of the miR-430 cluster found in chromosome 5 of the fSolSen1.1_lg genome assembly, and (**e**) localization of the miR-430 cluster in the mentioned chromosome.

**Figure 2 animals-13-02399-f002:**
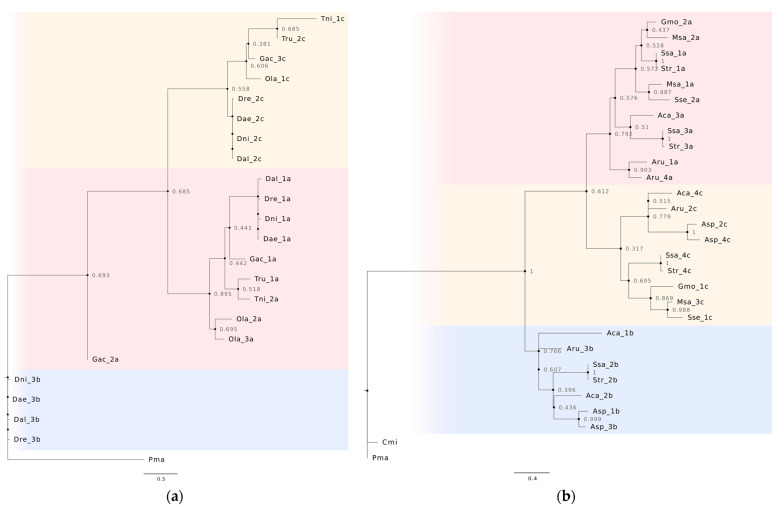
Phylogenetic tree obtained from a maximum likelihood analysis based on a Clustal Omega alignment of the miR-430 sequences found in the consensus repeat subunit of model species (**a**) and species of interest to fisheries and aquaculture (**b**), using the miR-430 sequence found in *Petromyzon marinus* as an outgroup. The numbers of the sequences are based on their position inside the repeat subunit, and the different variants have been classified based on their homology to the mature variants found in *Danio rerio*. The variants have been marked with different colours, with variant “a” in red, variant “b” in blue, and variant “c” in yellow. Numbers next to the nodes indicate the bootstrap values ranging from 0 to 1. Abbreviations: Aca = *Amia calva*, Aru = *Acipenser ruthenus*, Asp = *Atractosteus spatula*, Cmi = *Callorhinchus milii*, Dae = *Danio aesculapii*, Dal = *Danio albolineatus*, Dni = *Danio nigrofasciatus*, Dre = *Danio rerio*, Gmo = *Gadus morhua*, Gas = *Gasterosteus aculeatus*, Msa = *Micropterus salmoides*, Ola = *Oryzias latipes*, Pma = *Petromyzon marinus*, Ssa = *Salmo salar*, Str = *Salmo trutta*, Sse = *Solea senegalensis*, Tru = *Takifugu rubripes*, Tni = *Tetraodon nigroviridis*.

**Table 1 animals-13-02399-t001:** MiR-430 clusters. (**1**) Percentage of the genome occupied by the miR-430 clusters. (**2**) Consensus repeat subunit length in base pairs. (**3**) The number of miR-430 sequences in the consensus repeat subunit. (**4**) Variants to which these sequences correspond and their order, with the exception of *Petromyzon marinus*, as the only sequence found in this species does not group with any of the other variants.

Species	1	2	3	4
*Acipenser ruthenus*	0.0268	1060	4	a-c-b-a
*Amia calva*	0.0913	1481	4	b-b-a-c
*Atractosteus spatula*	0.0377	1443	4	b-c-b-c
*Danio rerio*	0.0473	1129	3	a-c-b
*Danio aesculapii*	0.0134	1132	3	a-c-b
*Danio albolineatus*	0.0365	1103	3	a-c-b
*Danio nigrofasciatus*	0.0780	1147	3	a-c-b
*Gadus morhua*	0.0146	251	2	c-a
*Gasterosteus aculeatus*	0.0469	846	3	a-a-c
*Micropterus salmoides*	0.0850	448	3	a-a-c
*Oryzias latipes*	0.0341	1438	3	c-a-a
*Salmo salar*	0.0132	1028	4	a-b-a-c
*Salmo trutta*	0.0119	1066	4	a-b-a-c
*Solea senegalensis*	0.0480	762	2	c-a
*Takifugu rubripes*	0.0222	741	2	a-c
*Tetraodon nigroviridis*	0.0238	796	2	c-a
*Petromyzon marinus*	0.0363	600	1	-

**Table 2 animals-13-02399-t002:** MiR-430 clusters in cichlids.

Species	Number of miR-430 Clusters	Consensus Repeat Subunit Length (bp)	Number of miR-430 Sequences in the Consensus Repeat Subunit	Seed Sequence *
*Amphilophus citrinellus*	1	546	1	A
*Astatotilapia stappersii*	1	935	3	A
*Ctenochromis horei*	3	227	1	U
		227	1	U
		377	1	A
*Cyprichromis leptosome*	2	425	1	U
		1128	3	A
*Haplochromis burtoni*	3	236	2	U
		500	2	U
		804	2	A
*Labeotropheus trewavasae*	2	288	2	A
		749	2	U
*Melanochromis auratus*	3	311	1	U
		323	1	U
		766	2	A
*Oreochromis niloticus*	2	637	3	U
		847	2	A
*Oreochromis korogwe*	2	256	1	U
		679	2	A
*Perissodus microlepis*	2	761	4	U
		1087	2	A
*Pseudocrenilabrus philander*	3	215	1	U
		313	1	U
		1026	3	A
*Pundamilia nyererei*	4	242	1	U
		247	1	A
		306	1	U
		426	1	A

* The seed sequence column marks whether the seed sequence of those miR-430 sequences is AAGUGCU (U) or AAGUGCA (A).

## Data Availability

Data available on request.

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
