# Peer review of "miR-430 microRNA Family in Fishes: Molecular Characterization and Evolution"

_animals, 2023, doi:10.3390/ani13152399_

Round 1

Reviewer 1 Report

In this manuscript, the authors analyzed the conservation of the mir-430 family in multiple fish species reporting cluster characterization, genes organization, and phylogeny of the variants analyses.  

Major Revision

In the tree could be reported the gain and losses of mir-340 variants An alignment among the mir-430 variants could help the reader in their identification.

Could this be the number of gene duplications related to whole genome duplication events? Could be interesting to verify if the number of copies is related to WGD (or TGD) or independent.

To investigate the Holostean genome, (for example Lepisosteus oculatus or Amia calva) could give interesting highlights on Ray-finned fish evolution

Minor Revision

Line 14 – “Early in the evolution of bony fishes” could be substituted with “early diverging species”

Line 14 -17 – “Certain variants have been related to specific functions” could be better rephrased)

Line 30 – when the authors intend to indicate several groups or species the form “fishes” should be used

Line 31 – the words already present in the title should be not present also as Keywords, so I suggest substituting the words “fish”, “mir-430 family” and “evolution”

Line 35 – “Post-transcriptional genetic regulation” should be substituted with “post-transcriptional gene regulation”

Line 57-58 – Not clear Reformulate

Line 96 – define “distinctive evolution mode” and as for the other groups report the cichlid examined species

Line 134 – “Possible” should be substituted with "putative

In my opinion the paper is well written even if some issues should be clarity in the introduction section

Reviewer 2 Report

In this manuscript Jimenez-Ruiz and colleagues aim to characterize the presence of the mir-430 in different fish clades.

While I find the scientific question interesting, the way the analysis are performed and the results presented makes the manuscript very hard to read and make the results very hard to interpret and to contextualize.

Figures are overall poorly described and most of the sentences in the results section are not supported by either figure/tables or references.

Given the above, I cannot recommend the publication of this manuscript.

English is OK.

Reviewer 3 Report

The author described how microRNAs regulate gene expression, particularly the miR-430 family involvement in early fish development, and how it affects sexual differentiation and the metamorphosis of flatfish.  The results show that this family appeared in fish evolution early and is found in multiple copies in all species studied.  Three variants were found early in the evolution of bony fishes. So this knowledge is crucial for regulating different functions in fish species, which could help improve fish farming. Generally speaking, the manuscript is well-written and worth publication after revisions.

General comments

1. Double-check the use of grammar and punctuation marks. Consider consulting a native English speaker to modify the language

2. The authors should note that each abbreviation needs to be defined upon its first use.

3. There are different abbreviations formats; the authors should cross-check all abbreviations to see if they have been written correctly as they appeared the first time

4. Please cross-check that all the references cited in the text are included in the list (and vice versa), use the correct citation format, and ensure they meet our journal’s requirements.

5. Why are maternal mRNA genes usually organized in cluster repeats and in a single chromosome?

Specific comments

i.                    Line 11: The abbreviation “mir-430/Mir-430” should be changed to “miR-430” Please cross-check throughout the entire manuscript and consider changing it.

ii.                  Line 14-15: What are those certain variants, and what are their specific function?

iii.                Line 27-29: Rephrase “Furthermore, ……………..rearrangements” to “Furthermore, we have detected isolated instances of the miR-430 repeat subunit in some cases, which suggests that this microRNA family may be affected by DNA rearrangements.”

iv.                Line 65, 283, 303, 312: Change these citations “Hadzhiev et al. (2023), Houbaviy et al . (2005), Thatcher et al. (2008), Desvignes et al. (2021)” to the proper citation style as required by MDPI journals

v.                  Line 77-80: Rephrase this paragraph

vi.                Line 86: Change “in” to “at”

vii.              Line 170: Add the article “the” before the word “number.”

viii.            Line 226: Please expand subtitle 3.4 “Cichlids.”

ix.                Line 277: Change “found” to “find.”

x.                  Line 229: Change “sequence” to “sequences”

xi.                Line 236-239: Rephrase “In both………………………hairpin”

xii.              Line 252: Add citation to support this statement“Although…….species”

xiii.            Line 318-321: Add citation “In cichlids………...seed sequence”

xiv.            Line 331: Rewrite “mir290-295”

No.

Reviewer 4 Report

Dear editor,

Greeting!

On behave of review the manuscript “ mir-430 microRNA family in fish: molecular characterization and evolution ” by Claudio and co-authors. In this study, the authors characterized the tandem repeats organization of mir-430 in different fish species. The relevant research results provided information for studying fish evolution. However, there are many vague statements in the manuscript. These need to be clarified further.

Major Point:

1. The Introduction is not well-written. It contains references of questionable relevance, has several grammatical errors, and is poorly organized. Most importantly, it fails to explicitly state the hypotheses tested by this study. The authors are encouraged to explicitly address (1) why miRNAs are important in studying the evolution of species. (2) the recent advances of miRNA-430 in aquatic organisms, especially model animals.

2. In Supplementary Table1, miRNA-430 clusters loci and chromosome numbers are only analyzed on part of the fishes, whether these fish were representative or not.

Moderate editing of English language

Round 2

Reviewer 2 Report

I think the manuscript has been greatly improved from the previous version and its readability has significantly increased.

The only comment I have is that I would only add a "limitation of the study" paragraph discussing how short-read approaches might be a limiting step in the genome-wide identification of such repetitive sequences.

Author Response

Thank you for your suggestions, we have added the limitations paragraph to the end of the discussion (lines 364-377).

Reviewer 3 Report

 It is ok, and can be accepted.

Author Response

Thank you for your suggestions that have improved the quality of the article.

Reviewer 4 Report

Minor editing of English language required

Minor editing of English language required

Author Response

English grammar and punctuation has gone through another round of revision by an English translator and a native English speaker.